# Genetic Basis of Potato Tuber Defects and Identification of Heat-Tolerant Clones

**DOI:** 10.3390/plants13050616

**Published:** 2024-02-23

**Authors:** Sanjeev Gautam, Jeewan Pandey, Douglas C. Scheuring, Jeffrey W. Koym, M. Isabel Vales

**Affiliations:** 1Department of Horticultural Sciences, Texas A&M University, College Station, TX 77843, USA; sanjeevgautam-1984@tamu.edu (S.G.); yourjeewan@tamu.edu (J.P.); d-scheuring@tamu.edu (D.C.S.); 2Texas A&M AgriLife Research and Extension Center, Lubbock, TX 79403, USA; jeff.koym@ag.tamu.edu

**Keywords:** genome-wide association study, genomic selection, high temperature stress, *Solanum tuberosum*, tuber deformities, quality

## Abstract

Heat stress during the potato growing season reduces tuber marketable yield and quality. Tuber quality deterioration includes external (heat sprouts, chained tubers, knobs) and internal (vascular discoloration, hollow heart, internal heat necrosis) tuber defects, as well as a reduction in their specific gravity and increases in reducing sugars that result in suboptimal (darker) processed products (french fries and chips). Successfully cultivating potatoes under heat-stress conditions requires planting heat-tolerant varieties that can produce high yields of marketable tubers, few external and internal tuber defects, high specific gravity, and low reducing sugars (in the case of processing potatoes). Heat tolerance is a complex trait, and understanding its genetic basis will aid in developing heat-tolerant potato varieties. A panel of 217 diverse potato clones was evaluated for yield and quality attributes in Dalhart (2019 and 2020) and Springlake (2020 and 2021), Texas, and genotyped with the Infinium 22 K V3 Potato Array. A genome-wide association study was performed to identify genomic regions associated with heat-tolerance traits using the GWASpoly package. Quantitative trait loci were identified on chromosomes 1, 3, 4, 6, 8, and 11 for external defects and on chromosomes 1, 2, 3, 10, and 11 for internal defects. Yield-related quantitative trait loci were detected on chromosomes 1, 6, and 10 pertaining to the average tuber weight and tuber number per plant. Genomic-estimated breeding values were calculated using the StageWise package. Clones with low genomic-estimated breeding values for tuber defects were identified as donors of good traits to improve heat tolerance. The identified genomic regions associated with heat-tolerance attributes and the genomic-estimated breeding values will be helpful to develop new potato cultivars with enhanced heat tolerance in potatoes.

## 1. Introduction

Heat stress during the potato growing season is a major environmental factor that adversely affects the growth, development, and productivity of potatoes (*Solanum tuberosum*), which is one of the most important staple crops in the world [1,2]. The effects of heat stress on potato crops have been linked with external and internal tuber defects [1,3]. External tuber defects can range from shape deformations to the presence of knobs or irregular secondary growths called gemmations. Heat sprouts occur when tubers sprout in the field before harvesting due to high temperatures. Regular growth interruptions lead to the formation of several tubers connected by the same stolon called chained tubers. When a growing tuber splits and heals, leaving a lengthwise fissure, it is called a ‘growth crack’. Potatoes without any external deformities can still be affected by heat stress, and the defects are evident upon cutting tubers obtained from heat-stress production sites. Common internal tuber defects observed in tubers harvested from areas subjected to heat stress are brown center/hollow heart, vascular discoloration, and internal heat necrosis. Hollow heart is a physiological disorder in potatoes characterized by a cavity in the pith region of the tuber that often starts as a brown center. The presence of gray to black discoloration in the vascular area of a potato tuber has been termed vascular discoloration. Necrotic brown areas in tuber pith tissue, termed internal heat necrosis or internal brown spots, cause potato tubers, especially chipping types, to be unsuitable for processing [4]. External and internal tuber abnormalities significantly affect the marketability of potato tubers [3]. External and internal tuber defects are also accompanied by quality deterioration with reduction in specific gravity resulting in lower tuber dry matter and an increase in reducing sugars [2]. Tuber quality deterioration causes huge economic losses to growers and processors. Despite attempts to incorporate heat tolerance in potatoes, developing a potato variety with all the desired yield and quality attributes that can thrive in high temperatures has remained challenging. Identifying traits and molecular markers linked to potato heat tolerance could facilitate the selection of parental clones and the development of new heat-tolerant potato cultivars [5].

Evaluating and selecting potato clones under high-heat-stress conditions can lead to the development of germplasm that is tolerant to high temperatures, which can be valuable in addressing the effects of global warming/climate change and expanding potato cultivation to areas where they are not currently grown. Potato is not a common staple crop in tropical lowlands because temperatures are too high for tuber development and growth [6]. However, developing new heat-tolerant potato cultivars may allow future expansion of potato production areas into warmer areas of the tropics. One example of this is in Bangladesh, where a heat-tolerant variety (BARI Alu-73) was released after local evaluation for seven years [7]. This variety is adapted to the environmental conditions in Bangladesh and could serve as a model for developing heat-tolerant potato cultivars in other regions.

Developing potato cultivars capable of withstanding high temperatures in regions prone to heat stress requires collecting high-quality quantitative data for genetic analysis, gene identification, and genetic transfer [8]. Genomic information and the association of genes with agronomic traits in potatoes are limited and have led to slow genetic-based advances in potato breeding [9]. Using a diverse genotype panel (with a broad genetic background) increases the probability of identifying quantitative trait loci (QTLs) that can be used for marker-assisted selection (MAS) [10]. Genome-wide association study (GWAS) allows for the identification of genomic regions associated with various traits [11]. GWAS have been employed as a tool to understand the genetic basis of complex traits in various species, including potatoes, by analyzing diverse panels of individuals and identifying QTLs and candidate genes with high mapping resolution [11,12,13]. As global temperatures continue to rise due to climate change, understanding the genetic basis of thermotolerance in potatoes becomes crucial for developing heat-resistant varieties and ensuring sustainable potato production [14].

The genetic basis of heat tolerance has been sought through different molecular studies, including transcriptomics, metabolomics, and transgenics, and important genes have been identified. Potatoes’ important temperature-responsive genes/proteins include patatin [15] and sucrose synthase (SuSy). The superoxide dismutase gene family has also been found to be responsible for conferring heat tolerance in potatoes [16] along with other antioxidants like ascorbate peroxidase (APX) and nucleoside diphosphate kinase (NDPK2) [17]. Heat-shock protein genes like HSP17.7 [18] and Heat Shock Cognate 70 are reported to confer heat tolerance in potatoes [19,20]. Transcription factors like CAPSICUM ANNUUM PATHOGEN FREEZING TOLERANCE PROTEIN 1 (*CaPF1*) [21], C-REPEAT BINDING FACTOR 1 (*CBF3*) [22], and ETHYLENE RESPONSE FACTOR (*StERF94*) [23] have been associated with heat tolerance in potatoes through transgenic studies. Certain genes, such as SELF PRUNING 6A (*StSP6A*), TERMINAL FLOWER-1 (*StTFL1*), POTATO HOMEODOMAIN 1 (*StPOTH1*), BEL1-LIKE PROTEIN (*StBEL5*), micro-RNA (*miR172*), POTATO MADS BOX 1 (*POTM1*), CYCLING DOF FACTOR (*StCDF*), STIP1 HOMOLOGY AND U-BOX CONTAINING PROTEIN (*StTUB19* and *StTUB7*), ABA RESPONSIVE ELEMENT(ABRE) BINDING FACTOR (*StABF2* and *StABF4*), have been identified as positive regulators of tuberization and are potential candidates for overexpression to confer heat tolerance in potato clones. On the other hand, genes like PHYTOCHROME B (*StPHYB*), CONSTANS (*StCO*), SUCROSE TRANSPORTER (*StSUT4*), SELF PRUNING 5G (*StSP5G*), and RELATED TO APETALA (*StRAP1*) act as negative regulators and could be targeted for suppression or inhibition to achieve heat tolerance [24]. However, little is known about the genomic regions associated with temperature responses on yield reduction and quality deterioration in potatoes [25].

Although there has been documented variability in potato heat tolerance, only a few reports are available that identify any QTLs associated with tuber-defect traits. A few QTL studies have focused on specific traits, such as tuber yield [19], growth cracks [26], hollow heart [27], and internal heat necrosis [28], instead of a collection of defects in regard to heat tolerance. Trapero-Mozos et al. [19] utilized the 06H1 biparental diploid potato population (extensively studied and characterized) to pinpoint a genetic locus linked to tuber yield under normal and slightly higher temperature conditions. They used nodal cuttings to phenotype heat tolerance. They reported a QTL at linkage group 4a, and the candidate gene HSP70, conferring heat tolerance, was later confirmed [20]. Though the study focused on the tuberization signal and overall yield, the defects observed in the field were not phenotyped. QTL in linkage group 11a was detected for growth cracks in a population obtained from a cross with 12601ab1 and Stirling [26]. Five QTLs have been detected in potatoes on chromosomes 3, 5, 6, and 12 for hollow heart [27]. Similarly, nine QTLs have been mapped for internal heat necrosis in tubers on chromosomes 4, 5, 7, and 10, which explained 3.7 to 14.5% of the phenotypic variation [28]. The association of molecular markers and phenotypic variation in both cultivated and wild potato varieties through GWAS can provide a strong foundation for identifying single-nucleotide polymorphisms (SNPs) associated with abiotic stress like heat stress [29].

Genomic selection has the potential to accelerate breeding for a variety of traits, including those with complex genetic inheritance. In potato breeding, genomic selection can be especially useful in early clonal generations to predict and select traits that are difficult or expensive to measure, or those with low heritability [30]. The availability of higher-density SNPs that span the entire genome of many plants now allows for the prediction of breeding values by regressing phenotypic values on all available markers [31]. In potato breeding programs, multiple traits are influenced by both major and minor effect loci but lack causal markers. However, the success of genomic selection depends on factors such as the size and similarity of the training and validation populations, as well as the genetic makeup of the traits being investigated [30].

There is no potato seed production in Texas, and the commercial potato area is low (~6000 ha) compared to other USA states [32]. However, Texas has a south-to-north gradient in which potatoes are planted and harvested covering a wide period of time, and fresh produce has high market value. Among the challenges faced by potato production in Texas, heat is an important abiotic stress (drought is not considered directly since fields benefit from center pivot irrigation), followed by diseases, mainly potato virus Y (PVY). Despite these constraints, Texas still has the highest summer crop yields among summer-crop-producing states in the USA [33]. Heat stress is a constant aspect in the major Texas potato growing areas. Most of the potato production in Texas is in the Panhandle area that historically benefited from lower night temperatures during potato tuberization. However, the Northern parts of the country have been experiencing heat spells more frequently. While other USA regions are slowly considering breeding for heat tolerance, the Texas A&M Potato Breeding Program has been selecting potatoes clones under heat stress for many years as heat stress has been a regular constraint in its selection fields. Since screening and selection of potatoes under high temperatures in Texas has been practiced by the Texas A&M Program, it is plausible that Texas bred and selected materials carry heat-tolerance genes. These materials thus provide an opportunity to study the underlying genetic mechanism of heat tolerance in potatoes.

Considering the need to improve the adaptation of potatoes to warm environments worldwide, this study aimed to identify chromosomal locations associated with tuber-related traits affected by heat stress. Three main objectives of the study were to (i) evaluate potato clones from the Texas A&M Potato Breeding Program for external and internal tuber-defect traits under heat stress, (ii) identify the genomic regions associated with tuber-defect traits under heat stress, and (iii) identify clones with favorable genomic-estimated breeding values that can be advanced in the breeding program or used as parents to develop new heat-tolerant clones.

## 2. Results

### 2.1. Temperature Conditions

During the crop growing period, both the Dalhart and Springlake locations experienced periods of high maximum temperatures (Appendix A). However, greater temperature fluctuations were observed in Springlake during the growing period, including at planting (max. ~20 °C/min. ~5 °C) and harvest (max. ~35–40 °C/min. ~20–25 °C) versus Dalhart (max. ~30 °C/min. ~10 °C at planting to max. ~30–35 °C/min. ~15–20 °C at harvest). In Springlake, the temperatures continued rising from planting time until harvest time. On the other hand, in Dalhart, the temperatures reached their highest point around 60–70 days after planting and then gradually decreased for the remainder of the crop season.

### 2.2. Trait Distributions and Heritability

Significant interactions between clones and environments were observed for all the traits measured except the percentage of tubers with chained tubers (Table 1). Clones were found to significantly differ for the traits measured except for the percentage of tubers with heat sprouts and the percentage of tubers with internal heat necrosis (Table 1). Considering 2020 data, locations differed significantly for most of the traits except for the tuber number per plant, the percentage of tubers with growth cracks, internal heat necrosis, and external defects (Table 1). Years did not differ significantly except for in the percentage of tubers with internal defects in Springlake trials (Table 1). Significant variation was found among market group for most traits except for the percentage of tubers with chained tubers, the percentage of tubers with growth cracks, and the percentage of tubers with vascular discoloration (Table 2).

The phenotypic distribution of yield traits was normal or close to normal (average tuber weight, tuber number per plant); however, tuber internal and external defect traits were skewed towards low values (Appendix A) because most genotypes evaluated were advanced selections or varieties, and selection was applied to reduce tuber defects during the early stages of their breeding cycle. Despite having skewed distributions, there was variation for tuber-defect traits in the diversity panel. The broad sense heritabilities obtained ranged from 0 to 0.94. Low heritabilities or heritabilities near zero were observed in some traits likely due to a large environmental effect and/or low phenotypic variation (Table 3).

Pearson’s correlations (Figure 1) for combined environments were positive between total yield and yield without culls (r = 0.98; *p* < 0.001), knobs and external defects (r = 0.91; *p* < 0.001), vascular discoloration and internal defects (r = 0.64; *p* < 0.001). Negative correlations were observed between the average tuber weight and tuber number per plant (r = −0.71; *p* < 0.001), hollow heart and tuber number per plant (r= −0.21; *p*= 0.002), chained tubers and average tuber weight (r = −0.22; *p* = 0.001). No significant correlations were observed between combined external defects and internal defects. However, a positive correlation was observed between growth cracks and hollow heart (r = 0.26; *p* = 0.0001). Yield also showed a positive correlation with internal defects (r = 0.22; *p* = 0.001) but no significant correlation with external defects.

### 2.3. Linkage Disequilibrium and GWAS Analysis

Population structure, linkage disequilibrium (LD), and relatedness were previously studied on the same diversity panel [34]. The calculated genome-wide LD was low. From the shape of the curve, a 5–10 Mb window seemed appropriate to filter the most significant markers in the output. This indicates that a lower marker density can be sufficient for association studies. In general, QTLs with small effects were observed.

The filtered SNP markers provided genome-wide coverage along the 12 chromosomes (Appendix A). No significant marker-trait associations were obtained for total yield, yield without culls, and yield of culls; however, three QTLs were detected for the yield-related traits of average tuber weight and average tuber number/plant (Table 4 and Figure 2). One QTL on chromosome 6 was detected for average tuber weight explaining 1.3% variation in the trait, while two QTLs were detected for average tuber number on chromosomes 1 and 10, explaining 0.7% and 0.9% variation in the trait. No QTLs were detected when all the external defects (heat sprouts, chained tubers, knobs, and growth cracks) and the internal defects (hollow heart, vascular discoloration, and internal heat necrosis) were combined. However, dissecting external and internal tuber defects into specific defect categories allowed for the detection of significant QTLs (Table 4 and Figure 2).

Significant SNP marker-trait association explained 0.5% to 2.0% of the phenotypic variation for the traits measured (Table 4). Two QTLs were identified on chromosomes 4 and 6, explaining around 1.5% each for the phenotypic variance in knobs. QTLs for chained tubers were found on chromosomes 1, 4, and 8, explaining 1.4% to 1.8% of phenotypic variation (Table 4 and Figure 2). For growth cracks, four QTLs were identified on chromosomes 3, 4, 6, and 11 explaining 1.0% to 1.9% of phenotypic variance. Four QTLs were detected for each hollow heart and internal heat necrosis. For hollow heart, 0.5% to 2.0% variation in the phenotype was explained with the QTLs detected in chromosomes 1, 2, 3, and 11. The QTLs for internal heat necrosis were detected on chromosomes 1, 3, 10, and 11, explaining 0.5% to 1.5% variation in the phenotype. No significant QTLs were detected for heat sprouts in the external defects category and vascular discoloration in the internal defects category. The annotated genes within the 250 kbp range of the SNP at QTL peaks detected have been listed in Appendix A. Recent similar GWAS studies have used different genomic region sizes—200 Kb [35], 100 Kb [36] and 600 Kb [37] to pinpoint the putative candidate genes.

### 2.4. Genomic Selection

Genomic-estimated breeding values for yield-related traits and different categories of external and internal defects were obtained using the StageWise package. The genomic-estimated breeding values ranged from 14.04 to 45.48 for total yield, 13.14 to 41.35 for yield without culls, and 1.04 to 5.23 for culls yield (Appendix A). Similarly, GEBVs ranged from 20.58 to 203.5 for average tuber weight, 3.91 to 14.31 for average tuber number/plant, 0.16 to 0.84 for the percentage of heat sprouts, 0.82 to 10.74 for the percentage of knobs, 0.13 to 3.53 for growth cracks, and 1.74 to 10.60 for the percentage of tubers affected with external defects. Similarly, the breeding values ranged from 0 to 10.68 for the percentage of hollow heart, 2 to 5 for the percentage of vascular discoloration, and 0.54 to 14.13 for the percentage of tubers affected with internal defects (Appendix A). The predicted reliability ranged from 0 in the case of chained tubers to over 0.8 in the case of average tuber weight (Figure 3). The highest GEBV for yield and yield without culls was found for AOTX98152-3Ru, while the lowest GEBVs were observed in ATX91322-2Y/Y. ATTX95490-2W had the highest GEBV for cull yield, whereas NDTX4828-2R showed the lowest GEBV in this category. ATX84706-2Ru had the highest GEBV for average tuber weight, and the lowest GEBV was found in ATX91322-2Y/Y. ATX08181-5Y/Y was found to have the highest GEBV for the average tuber number, while ATX87184-2Ru was found to have the lowest GEBV for the same trait (Table 5). Clones ATTX95490-2W and AOTX95309-1W showed high GEBVs for heat sprouts, while AOR07781-2 showed low GEBVs for heat sprouts. Russet Burbank had the second highest GEBV after COTX89044-1Ru for knobs, while clones like NDTX5003-2R, NDTX5067-2R, and NDTX0550169-1R showed the lowest GEBVs for knobs. Higher GEBVs for growth cracks were found in clones COTX09089-1Ru, and ATX843378-6Ru; however, lower GEBVs for growth cracks were observed in clones-ATX9202-3Ru and ATX08181-5Y/Y (Table 6). TX1475-3W, NDTX8773-4Ru were the clones with the highest GEBVs for hollow heart however, clones like NDTX059775-1W and AORTX09037-1W/Y were among the clones with lowest GEBVs for hollow heart. Atlantic had the highest GEBV for internal heat necrosis, whereas several red-skinned clones such as NDTX5003-2R, NDTX5067-2R, and NDTX0550169-1R had the lowest GEBVs for internal heat necrosis. TX090403-15W, COTX90046-1W were among the ones that showed highest GEBVs for vascular discoloration, whereas AOR07781-2, AOTX02136-1Ru, and COTX08365F-3P/P were among the clones that possessed the lowest GEBVs for vascular discoloration (Table 7). When the overall external defects are considered, clones like COTX89044-1Ru and Russet Burbank were found to possess the highest GEBV for external defects, and on the contrary clones like TX11461-2W and Rio Rojo had low GEBVs for external defects (Table 6). Similarly, the clones with the highest GEBVs for overall internal defects were TX1475-3W and NDTX8773-4Ru, whereas the lowest GEBVs were observed in NDTX059828-2W and ATTX98493-2P/P (Table 7). Cross-validation was performed by masking the phenotypic values of clones selected in Texas, but derived from crosses made in Aberdeen, ID (86 clones), and comparing the predicted phenotypes with the observed. It was generally observed that the predictions without phenotypes (marker-based selection—MBS) showed lower reliability than the predictions including observed phenotypes (marker-assisted selection—MAS) (Appendix A). A Z-score-based multi-trait-selection index facilitated the selection of superior clones (devoid of tuber defects) that are recommended as parents or for advancement in the breeding program (Appendix A). The examples of a few selected clones (based on WMIS) for each market group are AOR07781-2 and Vanguard Russet (Russet); TX11461-2W and TX12484-4W (Chip); ATTX98448-6R/Y and ATTX00289-5R/Y (Red); NDTX059886S-1Y/Y and NDTX071217CB-1W/Y(Yellow); NDTX081618-1P/P and COTX08039-1P/P* (Purple).

## 3. Discussion

Dalhart and Springlake, Texas, were selected as heat-stress screening sites to assess the effect of heat stress on potato yield, as well as on external and internal tuber defects. Both locations experience maximum temperature conditions (>25 °C) for most of the crop growing period and regularly reach above 40 °C for several days. These locations can be classified as heat-stress sites for potatoes based on the prevailing temperatures during the crop period, which align with other locations previously used to study heat stress on potatoes. Temperatures above 20 °C for 57% and 69% of the vegetative period have been reported as stressful environments [38], since potatoes need cooler nights to tuberize properly. Fernandes Filho et al. [39] have reported a maximum temperature around 30 °C as heat-stress environment for screening potatoes for heat stress. Al Mahmud et al. [40] found that temperatures reaching around 25 °C before the harvest resulted in heat stress on potatoes. A threshold temperature of 25 °C was used to assess the impact of heat stress on potatoes which defined a hot spell as two or more consecutive days with maximum temperatures at or above the threshold [41]. When comparing with previously published reports, Dalhart and Springlake, TX could be regarded as sites to evaluate prolonged heat stress on potatoes and could be used to identify heat-tolerant clones that withstand sporadic and potentially recurrent heat stress in other potato-growing regions in the USA and in other Northern Hemisphere areas.

The sensitivity of potatoes to heat stress has been associated with specific characteristics. Reduction in tuber yield is the most commonly reported trait. Under high temperatures, the majority of photoassimilates are directed towards plant shoots rather than the tubers, which can negatively impact the overall yield [42]. Lambert et al. [43] found that growing potatoes under high temperatures can lead to a reduction in both the number and weight of tubers, resulting in significant decreases in yield (up to 58%) and a reduction in the percentage of large tubers (by up to 25%). However, reports including different tuber defects like external and internal disorders, are rare. Internal defects such as hollow heart, vascular discoloration, stem ends, and internal heat necrosis can occur even in potatoes without external deformities. For example, Atlantic is a high-yielding chip variety with a low percentage of external defects, but internal heat necrosis is a major issue. This shows the importance of evaluating the effect of heat stress on not only yield parameters, but also on external as well as internal tuber defects.

Correlations between traits were calculated to gain insights into these complex traits and guide breeding efforts (Figure 1). Associations between genome-wide molecular markers and traits (GWAS) helped in understanding the genetic basis of traits. In this study, no significant correlation between yield and external defects indicates that yield could be increased without increasing external defects in potatoes. However, a positive correlation of yield with internal defects implies that the internal quality of potatoes might be affected by yield increase. No significant correlations between external defects and internal defects suggest that potatoes should be selected against external and internal defects separately. The positive correlation between hollow heart and growth cracks and also between hollow heart and internal heat necrosis indicate that their occurrence under heat stress might follow similar pathways. The negative correlation between average tuber weight and tuber number was expected. Several researchers have reported similar relationships [34,44]. QTLs for correlated traits did not overlap (Figure 2). The yield-related traits represent components of yield, and the tuber-defect traits are also components of external and internal tuber defects.

QTLs were not detected for total yield, yield without culls, and yield of culls in the present study. However, three QTLs associated with yield-related traits—average tuber weight and average tuber number per plant were detected (Table 4). The genomic region detected on chromosome 10 for the tuber number was also earlier reported to be associated with the tuber-number trait [45]. Also, the same QTL was found in earlier study to be associated with tuber shape [11,13]. A QTL detected on chromosome 6 for average tuber weight lies around 8 Mb from the QTL peak detected earlier for tuber weight [46]. Also, QTLs on chromosome 6 for tuber weight were detected in other studies [46,47]. When studying polygenic traits like yield, with complex architecture, the success of GWAS using low-density SNP array is comparatively lower than when targeting monogenic traits, which are controlled by large-effect QTLs [12]. However, there are reports where QTLs on yield-related traits have been reported. According to Caraza et al. (2016) [48], the most relevant SNP markers associated with marketable tubers were identified on chromosomes 5, 6, 11, and 12. In particular, under short-day conditions, SolCAP_c2_6000 and SolCAP_c2_20947 were identified on Chromosome 11, while SolCAP_c1_8002 and SolCAP_c2_34762 were identified on Chromosome 12. Under long-day conditions, highly associated markers were found on Chromosome 5 (SolCAP_c2_50302) and Chromosome 6 (SolCAP_c2_25926).

Detection of QTLs for all traits was not possible with current resources (marker density, population size, statistical software, and the complex nature of traits). Increasing population size, as well as marker density often improves QTL detection using GWAS in potatoes [12]. The utilization of marker-dense platforms in GWAS models allows for improved correction of varying degrees of relatedness and facilitates the identification of genomic regions that have been influenced by targeted breeding efforts, ultimately aiding in the development of cultivars with improved traits [49]. As previously reported by Pandey et al. (2021) [34], the examination of population structure and discriminant analysis of principal components in the panel revealed the presence of three sub-populations. GWASpoly considers population structure and relatedness, while finding marker-trait associations, to filter spurious associations out of the analysis. The present study used models on different polyploid gene actions, additive and dominant, through GWASpoly [12] for the detection of marker-trait associations.

The distribution of external and internal tuber defects showed skewed distribution toward low values. This was expected, since the breeding program applied early selection to reduce tuber defects. Despite that, phenotypic variation was still observed. The environmental conditions affected the expression of traits, as some defect categories were not observed in some environments, resulting in no phenotypic variation, and complicating the estimation of genetic parameters. Though the genotypes were exposed to high temperatures in all field trials, temperature fluctuations, photoperiod, relative humidity, wind, solar radiation, and other environmental differences can lead to differences in the expression of phenotypes leading to lower heritability and less power of association. Tuber-defect traits have been previously reported to have unclear or lacking heritable variation [50]. Skewed distributions on defects like internal heat necrosis [28] and hollow heart [27] have been reported. However, the presence of defects in several clones allowed us to carry out the association study to investigate genomic regions associated with the defects.

No QTLs were detected when all the external defects (heat sprouts, chained tubers, knobs, and growth cracks) were combined as a single trait ‘external defects’ and also when all internal defects (hollow heart, vascular discoloration, and internal heat necrosis) were combined as one single trait ‘internal defects’. However, when the external and internal tuber defects were dissected by category, QTLs were detected for most traits, except for tuber heat sprouts in the external defects category and tuber vascular discoloration in the internal defects category. The detection of QTLs for yield-contributing traits instead of overall yield traits in this study is similar to the detection of QTLs for separate tuber-defect traits instead of cumulative external or cumulative internal defect traits, indicating that when complex traits are dissected, it is easier to study their genetic basis. Tsutsumi-Morita et al. [51] also reported this finding.

The genetic basis of second growths or knobs is scarce [50]. We identified QTLs on chromosome 4 (SNP at peak: solcap_snp_c2_12947) at 69.18 Mb and chromosome 6 (SNP at peak: solcap_snp_c2_32923) at 31.74 Mb, explaining around 1.6% and 1.3%, respectively, of the phenotypic variance in knobs. Ubiquitin [52], cytochrome P450 [53], and methyltransferases [53] are some of the putative candidate genes near the SNP peak at chromosome 4 for knobs which could be involved in knob formation in potatoes. QTLs for knobs have not been reported previously. Only a few reports are available on genes or proteins responsible for knobs in potatoes, which include the ATP/ADP translocator (ANT or Adenine nucleotide translocator) protein, localized in the inner membrane of plastids [54], and Carotenoid cleavage dioxygenases [55].

For growth cracks, 1.0% to 1.9% of the phenotypic variance was explained due to QTLs identified on chromosome 3 (SNP at peak: PotVar0095359) at 38.7 Mb, chromosome 4 (SNP at peak: PotVar0087270) at 66.21 Mb, chromosome 6 (SNP at peak: PotVar0040975) at 55.57 Mb, and chromosome 11(SNP at peak: solcap_snp_c1_15802) at 42.9 Mb. Bradshaw et al. (2008) [26] detected a QTL on chromosome 11 for growth cracks, explaining a 5% variation in the phenotype. Candidate genes within 250 Kb distance from the SNP peaks found in our study suggest thecomplex nature of the trait, as several of these genes are reported to be involved in stress-related phenomena. Examples of the stress-related genes within that region include the Early responsive to dehydration protein [56], Homeobox-leucine zipper protein (*HAT22*) [57], and leucine-rich-repeat receptor-like protein kinase (*StLRPK1*) [58].

QTLs linked with chained tubers have not been reported earlier. However, a few studies reported tuber deformities similar to chained tubers in potatoes being linked to genes like Carotenoid cleavage dioxygenase4 (CCD4) [59] and Carotenoid cleavage dioxygenase8 (CCD8) [55]. The chained tubers’ phenotype has not been measured as a separate trait and has been, instead, often associated with dormancy [53]. Three QTLs were found for chained tubers on chromosomes 1, 4, and 8 explaining from 1.4% to 1.8% of the phenotypic variation in the current study (Table 4). A potential candidate gene in the genomic region for QTL (SNP peak-PotVar0086820) at 43.1 Mb on chromosome 8 for chained tubers is Pentatricopeptide repeat-containing protein (PGSC0003DMG400014513) reported to be involved in abiotic stress like drought [60].

Four QTLs were detected for internal heat necrosis. These QTLs were detected on chromosomes 1, 3, 10, and 11 explaining from 0.5% to 1.5% variation in the phenotype. Internal heat necrosis has been reported to occur in response to high temperatures during the growing season [4]. The first attempt to identify genomic regions for internal heat necrosis provided significant QTLs on chromosomes 4, 5, 7, and 10 on the tetraploid population B2721 obtained by crossing the internal heat necrosis susceptible variety Atlantic and the resistant clone B1829-5 [28]. In a follow-up study using the same population, QTLs for internal heat necrosis were identified on chromosomes 1, 5, 9, and 12, explaining 28.2% variation in internal heat necrosis [61]. The differences in the studies have been attributed to improved genome coverage by markers in the later study, along with the use of codominant markers in the second study. The genomic regions found on chromosomes 1 and 10 in our study did not coincide with the previously reported regions on the same chromosomes for internal heat necrosis.

For hollow heart, between 0.5% and 2.0% variation in the phenotype of hollow heart was explained through the QTLs detected in chromosomes 1, 2, 3, and 11. QTLs for hollow heart were previously reported on chromosomes 3, 5, 6, and 12 [27]. The QTL on chromosome 3 does not seem to coincide with the previously reported region on the same chromosome; however, it lies 2 Mb apart. The other three QTLs have not been previously reported. QTL detection of hollow heart is affected by the environment [27] and genetic background [27,62], since both have effects on the expression of a trait which is the result of the genes responsible. Potential candidate genes in the vicinity of SNP peaks for hollow heart associated with heat and drought stress in potatoes include, Ubiquitin-protein ligase [52,63], Laccase [64], BRASSINOSTEROID INSENSITIVE 1-associated receptor kinase [65], Mads box protein [63], and Zinc finger protein [63].

The main obstacle in breeding for tolerance to abiotic stress, including heat stress, lies in the accurate phenotyping process, which demands precise measurements and the utilization of relevant experimental conditions [66]. Detection of tuber defects in potatoes is only possible in harsh environments; however, in current potato-growing regions, the goal is to avoid extreme environments. Thus, obtaining phenotypes of defects under field conditions is extremely hard as the controlled environment conditions are quite cumbersome in the case of studies involving a large number of potato genotypes. Phenotyping for tuber defects is extremely difficult as the environment prevailing during the crop period determines the level of its expression. However, for this study, consistent high temperatures during the crop period were observed. Proper field conditions for the expression of the phenotype are required for GWAS.

To the best of our knowledge, this study represents the first attempt to identify genomic regions associated with tuber-defect traits under field heat stress, serving as a valuable reference for future studies. The detection of multiple QTLs that explain low phenotypic variation suggests that the traits are governed by multiple genes with very small effects [67]. Using a selected panel of clones with low phenotypic variation has likely limited the QTL detection and explains the low percentage of the phenotypic variance explained. In any case, the existing variation still allowed for detecting significant QTLs for most external and internal tuber categories. Achieving consistent results in identifying marker-trait associations in different population of the same species poses a significant challenge in terms of reproducibility [68].

Building a mapping population, segregating for specific defects, and evaluating it under heat-stress environments might help decipher the underlying genetics with more precision. Also, phenotyping and genotyping larger and more diverse populations [29] under a controlled and consistent heat-stress environment, allowing the defect traits to be expressed, could uncover genomic regions not detected in this study. The validation of significant QTLs in follow-up studies involving normal and heat-stress conditions, using transcriptomics, metabolomics, transgenics, as well as gene editing for contrasting genotypes for tuber-defect traits, could be completed. QTL detection for traits associated with heat tolerance has served an academic purpose. Considering the variation explained by the marker trait associations and the heritability of yield and tuber-defect traits, it is certain that GWAS has missed many small-effect loci. Thus, For practical breeding purposes, genomic selection has more value.

Genomic selection is a breeding approach that generates genome-estimated breeding values (GEBVs) for an inference population based on a reference population that has been both phenotyped and genotyped. To predict the performance of clones in the field, marker information is required for the inference population. The accuracy of the prediction is assessed by determining the correlation between the predicted phenotypes and the real phenotypic outcomes. For genomic selection to be effective in predicting the performance of lines with yet-to-be-observed phenotypes, a strong correlation between the predicted and real values is expected [31]. One of the main advantages of genomic selection in plant breeding is that it allows the selection of preferred clones for breeding, even if their phenotypes have not yet been observed.

The accuracy of the prediction methods can be compared through cross-validation, either by using another population or a subset of the same population. The graphs of predictions between MAS and MBS show that MAS reliability was consistently higher than MBS reliability across all individuals in almost all of the traits measured (Appendix A). This suggests that incorporating phenotypic data has improved the accuracy of predictions compared to relying solely on genetic markers. Since the traits measured are complex and the panel is not quite exhaustive, the predictions of an individual’s performance are not captured using genetic markers alone, emphasizing the importance of phenotypic data measurement before predictions are made highly accurate.

The genomic-estimated breeding values obtained using StageWise allowed for the selection of parents. Clones with high breeding values of defects should be avoided, while clones with low breeding values for defects should be selected as parents for the breeding program to obtain progeny with low levels of external and internal defects. The boxplot of predicted reliability indicates that traits like knobs, growth cracks, and hollow heart can be targeted for improvement. However, traits like heat sprouts, chained tubers, vascular discoloration, and internal heat necrosis must be further explored before being integrated into genomic selection.

A weighted multi-trait-selection index (WMIS) incorporating yield, external and internal tuber defects, processing quality, and disease/pest resistance should be used by breeders to select clones as parents or to advance them in the breeding program. However, correlations between traits and selection criteria for each market group should be considered when defining the list of traits included in the index and their respective weights. Assigning traits with their weights are subjective and under the discretion of each breeder [30]. These should match to the priority of growers, market demands, industry standards, and consumer preferences [69]. The simple weighted multi-trait-selection index calculated in the present study allowed for the selection of superior clones with few external and internal defects for different market groups as parents and/or for advancing clones in the breeding pipeline.

## 4. Materials and Methods

### 4.1. Plant Materials

A panel of 217 tetraploid clones was used for the study. The panel included selected advanced clones and varieties released by the Texas A&M Potato Breeding Program over several decades and reference varieties for various potato market groups. The collection comprised 31 chippers, 21 purples, 64 reds, 71 russets, and 30 yellows. The reference varieties used were Atlantic (Chipper); Russet Burbank (French fry processing); Russet Norkotah (Fresh market Russet); and Yukon Gold strain (Yellow-flesh fresh market). The panel was divided into three major sub-populations based on its structure and discriminant analysis [34].

### 4.2. Genotyping

Leaf tissue (50–80 mg) of in vitro cultured plantlets of each clone in the association panel was used for genomic DNA extraction with a DNeasy Plant Pro kit (Qiagen, Valencia, CA, USA). DNA quality and quantity were checked [34]. The DNA samples were genotyped using the Infinium 22K V3 Potato SNP Array on the Illumina iScan (Illumina Inc., San Diego, CA, USA) at Michigan State University and Neogen (Lansing, MI, USA). The SNP results were clustered using the Illumina GenomeStudio 2.0.4 software (Illumina, San Diego, CA, USA) based on the B allele (nulliplex = AAAA, simplex = AAAB, duplex = AABB, triplex = ABBB, and quadruplex = BBBB) using a custom tetraploid cluster file on the PolyGentrain polyploid module (Illumina, San Diego, CA, USA). The SNP genotype data were filtered to exclude low-quality monomorphic SNPs and loci with ≥10% missing data [34].

### 4.3. Field Experiment and Phenotyping

Field experiments were conducted in Texas: Dalhart (35°58′ N, 102°44′ W) during 2019 and 2020, while in Springlake (34°6′ N, 102°19′ W) they were conducted during 2020 and 2021. Planting, management, and harvesting of trials were aligned with commercial potato production in the corresponding areas. Trials were planted in March in Springlake and harvested in July (Appendix A). On the other hand, in Dalhart planting took place in May and harvest in September (Appendix A). The Dalhart location is 204 km north of the Springlake location and is at a higher altitude (1214 m vs. 1122 m above sea level). The average daily maximum and minimum temperatures were obtained online (https://www.ncdc.noaa.gov/cdo-web/, accessed on 12 December 2022) from the NOAA’s weather stations of Littlefield and Dalhart FAA airport, near the field sites. Each clone was planted in two replications (plots) consisting of 12 tuber seed pieces each. At planting, seed spacing was at 30 cm apart within rows and 70 cm between rows. The cultural practices were performed according to the grower’s practices (https://potato.tamu.edu/reports/, accessed on 11 August 2023). Fertilizers and irrigation were applied through a central pivot irrigation system. The plants haulms were mechanically destroyed a few days before harvesting. Harvest was completed using a two-row mechanical digger. The tubers were picked by hand into sacks and brought to a grading table for data recording.

The harvested tubers were graded according to standard protocol followed by the Texas A&M University potato breeding program (https://potato.tamu.edu/reports/, accessed on 11 August 2023). All harvested tubers from a plot were counted and weighed. Total yield is a measure of the weight of all the tubers in a plot expressed in Mg/ha. Cull tubers (tubers with external defects, insect/mechanical damage, greenheads, rots) were separated from non-culls. Yield without culls was recorded as the weight of all tubers minus the weight of culls. In addition, for this study, the cull tubers were further categorized into their prominent external tuber defects (Figure 4A). For Dalhart, 2019, data were not collected for external defects. Tubers with defects were categorized into knobs and growth cracks in Dalhart 2020, and into all four external defect categories (knobs, growth cracks, chained tubers, and heat sprouts) in Springlake trials (2020 and 2021). The number of tubers in each of the defect categories was recorded and expressed as a percentage of the total number of tubers. Internal tuber defects were measured in all four environments by cutting the ten-largest tubers per replication (Figure 4B) from the bud end to the stem end, and they were expressed as a percentage. The total percentage of external defects was calculated by adding all external tuber defects into one category, and they were divided by the total number of tubers in the plot. The total percent of internal tuber defects was calculated by adding individual internal tuber defects categories. The average tuber weight was obtained by dividing the total tuber weight by the total tuber count in the plot. The tuber number per plant was obtained by dividing the total tubers from a plot by the total plants in the plot before harvest. Phenotypic distributions were obtained from all traits, and a normality test (Shapiro–Wilk test) was performed.

### 4.4. Statistical Analysis

Yield and tuber-defect traits were analyzed separately and combined for four environments (location-year) using software META-R (Version 6.0) [70]. Best linear unbiased estimates (BLUEs) and heritability estimates were generated using the protocol implemented in META-R. Phenotypic correlations between the traits measured were calculated as Pearson’s correlations and visualized using JMP pro 17^®^ [71]. An analysis of variance across environments, locations, and years was carried out using JMP pro 17^®^. For a combined analysis across environments, clones were treated as fixed effects, whereas environments, replication within environments, and interaction between environments and clones were treated as random effects. To determine the effects of locations, Dalhart 2020 and Springlake 2020 data were used, and locations were treated as fixed effects. Years were treated as random effects and were evaluated separately for Springlake and Dalhart. For comparison among different market groups, combined across all environments, overall BLUE values were analyzed based on Tukey’s tests.

A total of 10,106 SNP markers were retained for analysis after the genotypic dataset was filtered [34]. The filtered SNP markers were used for GWAS together with the phenotypic dataset. Marker–trait association mapping was performed for yield and quality (external and internal tuber defects) traits using GWASpoly package (Version 2.12) [12]. The leave-one-chromosome-out (LOCO) method was implemented to account for population structure. Additive and dominant genetic models were tested for each trait. Bonferroni test was conducted for each trait to establish a LOD threshold corresponding to a genome-wide false-positive rate of 5%. The location of significant QTLs on different chromosomes was visualized with package chromoMap (Version 4.1.1) [72]. To identify putative candidate genes in QTL regions, genes were retrieved from the potato reference genome of *Solanum tuberosum* group Phureja DM1-3 PGSC v4.03 from the Spud database (http://solanaceae.plantbiology.msu.edu/, accessed on 5 June 2023). Genomic-estimated breeding values (GEBV) were calculated using the StageWise package (version 0.26) [73] which is based on a two-stage approach in R [74] using ASReml-R v4 [75] for REML estimation of variance components. The standardized weighted multi-trait-selection index (WMIS) (for selection of parents/advance clones) within each market group was calculated on Z values of GEBVs on yield and defects traits following a similar approach to the one described by Pandey et al. [30]. A simple WMIS for each market group was obtained by summing the product of Z values of GEBVs on the total yield and external and internal defects, keeping their associated trait weights equal. Positive weights are assigned for desirable traits (total yield, average tuber weight, tuber number per plant) and negative weights for undesirable traits (defect traits). The calculated WMIS was converted into a Z value—WMIS (Z). This will allow breeders to select among the top genotypes [highest WMIS (Z)] as potential parents (or to advance in the breeding pipeline) while avoiding the ones with low values.

## 5. Conclusions

Rather than using the total tuber external and total internal defects as traits, it is preferred to study individual components of external trait defects (knobs, chained tubers, growth cracks) and internal traits defects (hollow/heart, internal tuber necrosis, vascular discoloration) separately in order to understand the genetic basis of tuber deformities. A total of 17 QTLs related to various categories/types of external and internal tuber defects connected with heat stress on potatoes were identified along with three QTLs associated with yield-related traits. Genomic-estimated breeding values (GEBVs) allow you to choose parents or advanced clones that have low levels of external and internal defects under high-temperature conditions. GEBVs for internal and external tuber defects can be obtained based on molecular markers alone (MAB—marker-based breeding), and in this case no phenotyping under high-temperature conditions would be necessary; however, the GEBVs will be better (higher reliability) if the clones are genotyped and phenotyped (MAS: marker-assisted selection) under heat-stress conditions. Using a standardized weighted selection index for multiple traits is recommended for practical breeding purposes as it aids in the selection of multiple traits at the same time.

## Figures and Tables

**Figure 1 plants-13-00616-f001:**
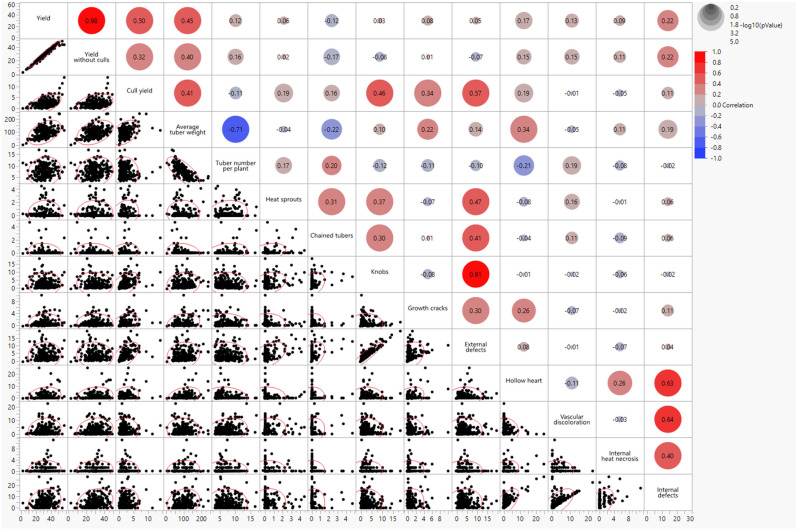
Pearson’s correlations for yield and tuber-defect traits measured in 217 potato genotypes across Texas environments [Dalhart (2019 and 2020) and Springlake (2020 and 2021)].

**Figure 2 plants-13-00616-f002:**
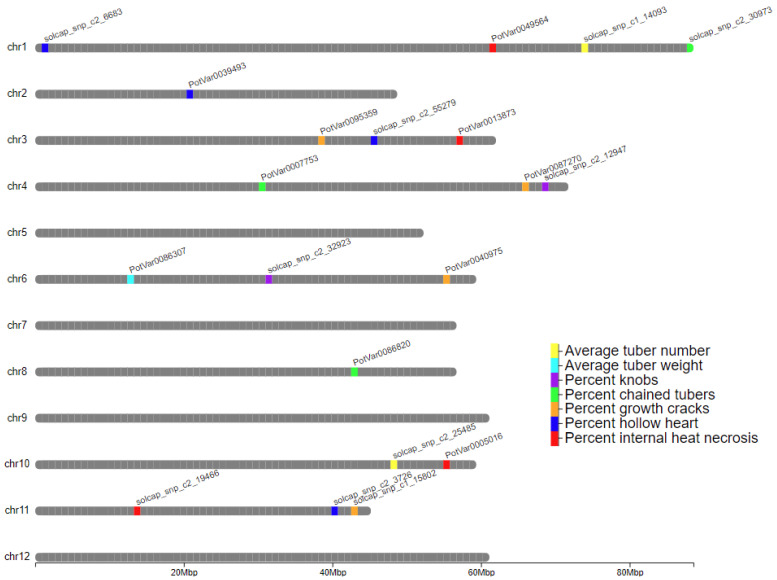
Location of significant QTLs on potato chromosomes for yield components and tuber external and internal defects based on evaluations of 217 potato genotypes across Texas environments [Dalhart (2019 and 2020) and Springlake (2020 and 2021)]. The QTLs are color-coded and the closest SNP to each peak is indicated.

**Figure 3 plants-13-00616-f003:**
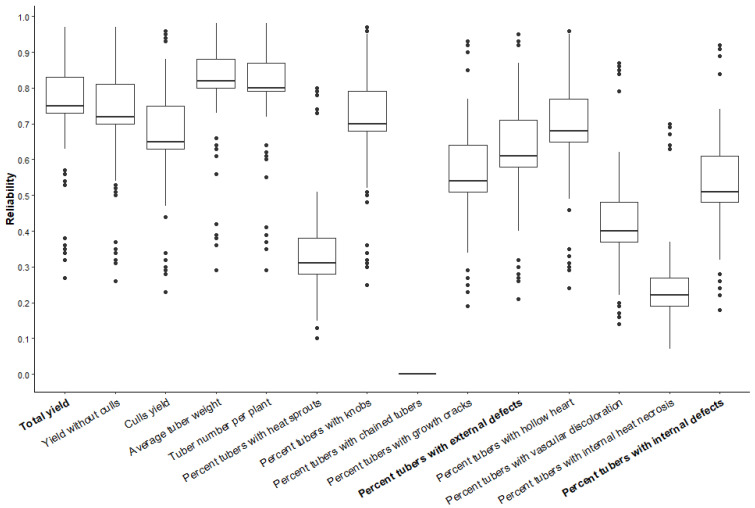
Boxplot of the predicted reliabilities (*y* axis) for the yield and tuber-defect traits (*x* axis) measured in 217 potato genotypes in Dalhart (2019 and 2020) and Springlake (2020 and 2021).

**Figure 4 plants-13-00616-f004:**
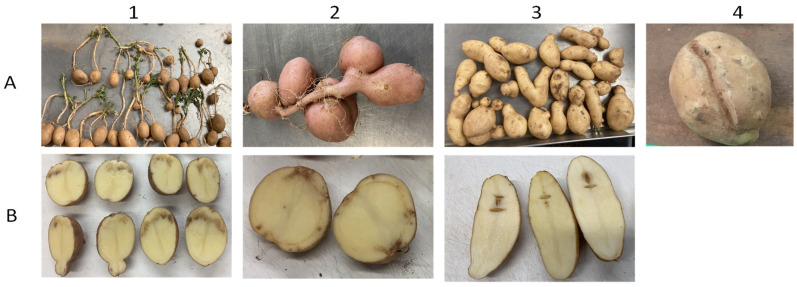
Phenotypes of potato tuber defects. Horizontal panel (**A**) represents external defects: (1) heat sprouts, (2) chained tubers, (3) knobs, and (4) growth cracks. Horizontal panel (**B**) represents internal defects: (1) internal heat necrosis, (2) vascular discoloration, and (3) hollow heart.

**Table 1 plants-13-00616-t001:** Analysis of variance to assess the effects of (A) environments, (B) locations, and (C) years on yield and tuber-defect traits evaluated in Texas potato growing environments: Dalhart (2019 and 2020) and Springlake (2020 and 2021).

Source of Variation	Total Yield	Yield without Culls	Culls Yield	Average Tuber Weight	Tuber Number per Plant	Percent Tubers with Heat Sprouts	Percent Tubers with Knobs	Percent Tubers with Chained Tubers	Percent Tubers with Growth Cracks	Percent Tubers with External Defects	Percent Tubers with Hollow Heart	Percent Tubers with Vascular Discoloration	Percent Tubers with Internal Heat Necrosis	Percent Tubers with Internal Defects
A. Environments (Dalhart 2019, 2020 and Springlake 2020, 2021)														
Clone	***	***	***	***	***	ns	***	***	***	***	***	***	ns	***
Env	ns	ns	ns	ns	ns	ns	ns	ns	ns	ns	ns	ns	ns	ns
Rep [Env]	ns	ns	ns	ns	ns	ns	ns	ns	ns	ns	ns	ns	ns	*
Clone*Env	***	***	***	***	***	**	***	ns	***	***	***	**	***	***
B. Locations (Dalhart 2020, Springlake 2020)														
Clone	***	***	ns	***	***	- ^&^	***	-	***	***	***	*	ns	ns
Location	***	***	***	***	ns	-	*	-	ns	ns	**	**	ns	***
Rep (Location)	ns	ns	ns	***	ns	-	ns	-	ns	ns	ns	ns	ns	ns
Clone*Location	***	***	***	***	**	-	**	-	ns	ns	***	**	ns	*
C. Years														
(i) Dalhart 2019, 2020														
Clone	***	***	***	***	***	-	-	-	-	-	***	***	ns	ns
Year	ns	ns	ns	ns	ns	-	-	-	-	-	ns	ns	ns	ns
Rep (Year)	ns	ns	ns	ns	ns	-	-	-	-	-	*	*	ns	ns
Clone*Year	***	***	***	***	***	-	-	-	-	-	***	***	ns	***
(ii) Springlake 2020, 2021														
Clone	***	***	***	***	***	ns	***	***	**	***	*	ns	ns	**
Year	ns	ns	ns	ns	ns	ns	ns	ns	ns	ns	ns	ns	ns	***
Rep (Year)	ns	ns	ns	ns	ns	ns	ns	ns	ns	ns	ns	ns	ns	***
Clone*Year	***	***	***	***	***	***	***	ns	***	***	***	***	***	***

*, **, ***—Statistically significant at *p* < 0.05, 0.01 and 0.001 respectively; ns—not significant; ^&^—Dash lines indicate that data for the traits were not collected.

**Table 2 plants-13-00616-t002:** Means (BLUEs) by market group for potatoes evaluated for yield and tuber-defect traits in Texas potato growing environments: Dalhart (2019 and 2020) and Springlake (2020 and 2021).

Market Group (No. of Clones)	Total Yield	Yield without Culls	Culls Yield	Average Tuber Weight	Tuber Number per Plant	Percent Tubers with Heat Sprouts	Percent Tubers with Knobs	Percent Tubers with Chained Tubers	Percent Tubers with Growth Cracks	Percent Tubers with External Defects	Percent Tubers with Hollow Heart	Percent Tubers with Vascular Discoloration	Percent Tubers with Internal Heat Necrosis	Percent Tubers with Internal Defects
	Mg/ha	Mg/ha	Mg/ha	g										
Chip (31)	35.29 A	32.95 A	2.33 AB	108.39 B	7.04 B	0.10 B	2.32 AB	0.09 A	0.62 A	3.08 AB	1.85 AB	3.37 A	1.37 A	6.58 A
Purple (21)	25.19 B	23.24 B	1.94 AB	68.67 C	8.15 AB	0.3 AB	4.33 A	0.26 A	0.30 A	4.99 AB	0.09 B	1.37 A	0.19 B	1.62 B
Red (64)	31.12 AB	29.64 A	1.48 B	74.77 C	9.19 A	0.25 B	1.69 B	0.16 A	0.95 A	2.91 B	0.35 AB	3.46 A	0.28 B	4.07 AB
Russet (71)	32.04 A	29.32 A	2.58 A	132.28 A	5.03 C	0.36 B	3.54 A	0.01 A	0.89 A	4.68 A	2.43 A	2.28 A	0.46 B	5.15 A
Yellow (30)	30.98 AB	28.95 AB	1.91 AB	69.60 C	9.40 A	0.83 A	2.93 AB	0.34 A	0.38 A	4.07 AB	1.43 AB	4.06 A	0.5 AB	6.32 A

Values followed by the same letters (within a column) were not significantly different based on Tukey’s tests (*p* < 0.005).

**Table 3 plants-13-00616-t003:** Broad-sense heritability for tuber-defect traits evaluated in Texas potato growing environments: Dalhart (2019 and 2020) and Springlake (2020 and 2021).

Environment	Total Yield	Yield without Culls	Culls Yield	Average Tuber Weight	Average Tuber Number per Plant	Percent Tubers with Heat Sprouts	Percent Tubers with Chained Tubers	Percent Tubers with Knobs	Percent Tubers with Growth Cracks	Percent Tubers with External Defects	Percent Tubers with Hollow Heart	Percent Tubers with Vascular Discoloration	Percent Tubers with Internal Heat Necrosis	Percent Tubers with Internal Defects
Dalhart 2019	0.66	0.68	0.56	0.83	0.72	- ^&^	-	-	-	-	0.00	0.40	0.00	0.29
Dalhart 2020	0.73	0.71	0.69	0.87	0.79	-	-	0.63	0.78	0.68	0.71	0.00	0.00	0.68
Springlake 2020	0.72	0.72	0.63	0.94	0.79	0.39	0.35	0.73	0.45	0.69	0.46	0.48	0.00	0.47
Springlake 2021	0.73	0.72	0.79	0.82	0.74	0.00 ^#^	0.21	0.66	0.52	0.63	0.00	0.60	0.54	0.60
Overall	0.61	0.59	0.45	0.88	0.82	-	0.19	0.50	0.30	0.46	0.14	0.31	-	0.25

^&^ Dash lines indicate that data for the traits were not collected. ^#^ Very low heritabilities or zero could be due to the low variability of the trait observed and/or a large environmental effect.

**Table 4 plants-13-00616-t004:** Marker-trait associations for potato tuber-defect traits evaluated in Texas: Dalhart (2019 and 2020) and Springlake (2020 and 2021).

Tuber Trait	Model	Threshold	SNP at QTL Peak	Chr.	Peak Position (bp)	Score [−log10(p)]	R^2^ (%)
Total yield	- ^&^	-	-	-	-	-	-
Average tuber weight	1-dom-alt	4.97	PotVar0086307	6	12,615,306	5.46	1.3
Average tuber number	1-dom-alt	4.97	solcap_snp_c1_14093	1	73,743,526	5.85	0.9
	1-dom-alt	4.97	solcap_snp_c2_25485	10	48,737,840	5.65	0.7
Percent tubers with external defects	- ^&^	-	-	-	-	-	-
Percentage knobs	1-dom-ref	5.13	solcap_snp_c2_12947	4	69,185,901	5.81	1.6
	1-dom-alt	5.03	solcap_snp_c2_32923	6	31,746,535	5.08	1.3
Percentage chained tubers	1-dom-ref	5.13	solcap_snp_c2_30973	1	88,322,772	6.44	1.7
	1-dom-ref	5.13	PotVar0007753	4	30,716,340	5.66	1.4
	additive	5.31	PotVar0086820	8	43,123,854	5.64	1.8
Percentage growth cracks	1-dom-alt	5.03	PotVar0095359	3	38,707,513	5.15	1.2
	1-dom-alt	5.03	PotVar0087270	4	66,211,523	6.12	1.2
	1-dom-alt	5.03	PotVar0040975	6	55,572,089	6.85	1.9
	1-dom-alt	5.03	solcap_snp_c1_15802	11	42,956,372	6.40	1.0
Percentage tubers with internal defects	- ^&^	-	-	-	-	-	-
Percentage hollow heart	1-dom-alt	5.03	solcap_snp_c2_6683	1	1,828,457	6.95	0.5
	additive	5.31	PotVar0039493	2	20,838,472	7.60	2.0
	1-dom-ref	5.13	solcap_snp_c2_55279	3	45,386,107	6.23	1.4
	1-dom-alt	5.03	solcap_snp_c2_3726	11	40,187,661	9.33	0.6
Percentage internal heat necrosis	1-dom-ref	5.13	PotVar0049564	1	62,154,569	5.87	0.5
	1-dom-ref	5.13	PotVar0013873	3	57,569,599	5.45	0.7
	1-dom-alt	5.03	PotVar0005016	10	55,841,256	6.06	0.7
	additive	5.31	solcap_snp_c2_19466	11	13,771,649	5.60	1.5

^&^ Dash lines indicate that no QTLs were detected.

**Table 5 plants-13-00616-t005:** Genomic-estimated breeding values (GEBV) of potato clones (highest—bold font, lowest—italics font, and reference varieties—underlined) for yield traits evaluated in Texas across environments [Dalhart (2019 and 2020) and Springlake (2020 and 2021)].

Clone	Total Yield	Clone	Yield without Culls	Clone	Culls Yield	Clone	Average Tuber Weight	Clone	Tuber Number per Plant
**AOTX98152-3Ru**	**45.48**	**AOTX98152-3Ru**	**41.35**	**ATTX95490-2W**	**5.23**	**ATX84706-2Ru**	**203.50**	**ATX08181-5Y/Y**	**14.31**
**TX13590-9Ru**	**45.42**	**TX13590-9Ru**	**41.32**	**MWTX2609-4Ru**	**4.55**	**ATX84378-6Ru**	**187.03**	**ATTX06246-1R**	**11.96**
**TXA549-1Ru**	**45.39**	**TXA549-1Ru**	**41.28**	**MWTX2609-2Ru**	**4.54**	**AOTX96216-1Ru**	**175.62**	**ATX05186-1R**	**11.53**
**ATTX95490-2W**	**45.33**	**White LaSoda**	**39.87**	**MWTX548-2Ru**	**4.51**	**AOTX96216-2Ru**	**175.32**	**ATTX05175s-1R/Y**	**11.27**
**White LaSoda**	**44.64**	**TX11461-2W**	**39.09**	**AOTX96216-1Ru**	**4.42**	**TX1475-3W**	**157.22**	**ATTX05186-2R**	**11.22**
**ATX84706-2Ru**	**43.93**	**AOR07781-2**	**38.81**	**ATX84378-6Ru**	**4.37**	**NDTX4930-5W**	**156.90**	**NDTX092237C-2P/W**	**11.20**
**AOR07781-2**	**43.15**	**AOTX91861-4R**	**38.77**	**AOTX96216-2Ru**	**4.30**	**AOTX98152-3Ru**	**154.99**	**ATTX98465-1R/Y**	**11.16**
**COTX08258-6Ru**	**42.07**	**TX1475-3W**	**38.73**	** Russet Burbank **	** 4.17 **	**TX13590-9Ru**	**153.77**	**ATX91322-2Y/Y**	**10.88**
**Vanguard Russet**	**41.56**	**ATX84706-2Ru**	**38.70**	**White LaSoda**	**3.79**	**TXA549-1Ru**	**153.68**	**ATX07305S-1Y/Y**	**10.83**
**ATTX98448-6R/Y**	**41.30**	**ATTX98448-6R/Y**	**38.25**	**ATX84706-2Ru**	**3.65**	**ATX87184-2Ru**	**153.25**	**COTX05249-3W/Y**	**10.61**
Atlantic	36.89	Atlantic	35.22	TXYG79	2.42	Russet Norkotah	129.06	Atlantic	6.91
TXYG79	34.50	TXYG79	31.79	Atlantic	2.16	Atlantic	115.09	TXYG79	6.54
Russet Norkotah	29.56	Russet Norkotah	27.53	Russet Norkotah	1.57	TXYG79	109.10	Russet Burbank	5.95
Russet Burbank	29.05	Russet Burbank	24.42	*JTTX75/2003EH-1Yre/Y*	*1.38*	Russet Burbank	94.17	Russet Norkotah	4.46
*COTX08365F-3P/P*	*22.95*	*TX10437-9Pyspl/Y*	*21.00*	*ATTX05175s-1R/Y*	*1.35*	*PTTX05PG07-1W*	*50.95*	*AOTX98137-1Ru*	*4.46*
*NDTX091886-3P/P*	*22.09*	*NDTX050169-1R*	*20.96*	*NDTX092238Cs-1P/W*	*1.35*	*ATX07305S-1Y/Y*	*50.62*	*TXNS106*	*4.46*
*ATTX98514-1R/Y*	*21.58*	*ATTX98514-1R/Y*	*20.26*	*COTX08039-1P/P**	*1.32*	*PORTX03PG25-2R/R*	*50.42*	*AOTX95295-3Ru*	*4.45*
*ATTX98500-3P/Y*	*21.44*	*COTX08365F-3P/P*	*20.24*	*NDTX059828-2W*	*1.32*	*ATTX05175s-1R/Y*	*49.48*	*AOTX95265-4Ru*	*4.43*
*NDTX050169-1R*	*21.23*	*ATTX98500-3P/Y*	*19.20*	*TX12474-1P/R*	*1.31*	*COTX10226-1Wpe/Y*	*48.55*	*ATX9117-1Ru*	*4.32*
*PORTX03PG25-2R/R*	*21.00*	*PORTX03PG25-2R/R*	*19.10*	*COTX94218-1R*	*1.25*	*NDTX050169-1R*	*45.53*	*AOTX96216-2Ru*	*4.18*
*TX12474-1P/R*	*17.69*	*TX12474-1P/R*	*17.13*	*Rio Rojo*	*1.20*	*ATX08181-5Y/Y*	*44.01*	*AOTX96216-1Ru*	*4.14*
*CO112-F2-2P/P*	*17.43*	*CO112-F2-2P/P*	*16.59*	*NDTX050169-1R*	*1.08*	*TX12474-1P/R*	*42.4*	*ATX84378-6Ru*	*4.05*
*PTTX05PG07-1W*	*17.16*	*PTTX05PG07-1W*	*16.07*	*NDTX4828-2R*	*1.04*	*CO112-F2-2P/P*	*36.47*	*ATX87184-2Ru*	*3.91*
*ATX91322-2Y/Y*	*14.04*	*ATX91322-2Y/Y*	*13.14*			*ATX91322-2Y/Y*	*20.58*		

Clones with bold font represent having highest GEBVs; clones underlined are reference varieties; and clones with italic font are the clones with lowest GEBVs.

**Table 6 plants-13-00616-t006:** Genomic-estimated breeding values (GEBVs) of potato clones (highest—bold font, lowest—italic font, and reference varieties—underlined) for external tuber-defect traits evaluated in Texas across environments [Dalhart (2019 and 2020) and Springlake (2020 and 2021)].

Clone	Percent Heat Sprouts	Clone	Percent Knobs	Clone	Percent Growth Cracks	Clone	Percent External Defects
**ATTX95490-2W**	**0.84**	**COTX89044-1Ru**	**10.74**	**COTX09089-1Ru**	**3.53**	**COTX89044-1Ru**	**10.60**
**AOTX95309-1W**	**0.80**	** Russet Burbank **	** 8.41 **	**ATX84378-6Ru**	**2.88**	** Russet Burbank **	** 9.13 **
**ATX85404-8W**	**0.80**	**COTX08365F-3P/P**	**6.87**	**AOTX96216-1Ru**	**2.78**	**ATX97147-4Ru**	**8.97**
**COTX03079-1W/W**	**0.68**	**COTX04050s-1P/P**	**6.85**	**AOTX96216-2Ru**	**2.73**	**COTX04050s-1P/P**	**8.36**
**NDTX092237C-2P/W**	**0.68**	**MWTX2609-4Ru**	**6.84**	**ATX84706-2Ru**	**2.47**	**COTX08365F-3P/P**	**8.13**
**White LaSoda**	**0.64**	**MWTX2609-2Ru**	**6.81**	**COTX01403-4R/Y**	**2.46**	**ATX84378-6Ru**	**7.52**
**ATX84378-6Ru**	**0.53**	**MWTX548-2Ru**	**6.75**	**NDTX731-1R**	**2.31**	**MWTX2609-4Ru**	**7.52**
**ATTX00289-4W**	**0.52**	**ATTX98500-3P/Y**	**5.64**	**NDTX7590-3R**	**2.18**	**MWTX2609-2Ru**	**7.51**
**NDTX081648CB-4W**	**0.52**	**ATTX98493-2P/P**	**5.43**	**ATX9130-1Ru**	**1.97**	**MWTX548-2Ru**	**7.46**
**Sierra GoldTM**	**0.52**	**ATTX10265-4R/Y**	**5.37**	**TX05249-10W**	**1.80**	**AOTX96216-1Ru**	**7.38**
Russet Burbank	0.51	TXYG79	2.90	Atlantic	0.95	TXYG79	3.94
Atlantic	0.46	Russet Norkotah	2.53	TXYG79	0.67	Atlantic	3.50
TXYG79	0.40	Atlantic	1.98	Russet Burbank	0.28	Russet Norkotah	2.73
* Russet Norkotah *	0.21	*NDTX059759-3R/Y Pinto*	*1.26*	Russet Norkotah	0.25	*ATTX98466-5R/WR*	*2.09*
*Russet Norkotah 102*	*0.21*	*TX12474-1P/R*	*1.22*	*TX12474-1P/R*	*0.24*	*BTX2103-1R/Y*	*1.99*
*Russet Norkotah 112*	*0.21*	*Rio Rojo*	*1.21*	*ATTX05175s-1R/Y*	*0.23*	*COTX08039-1P/P**	*1.99*
*Russet Norkotah 223*	*0.21*	*BTX2103-1R/Y*	*1.20*	*ATX03564-1Y/Y*	*0.23*	*NDTX050169-1R*	*1.99*
*Russet Norkotah 278*	*0.21*	*COTX94218-1R*	*1.07*	*ATX03496-3Y/Y*	*0.22*	*COTX94218-1R*	*1.88*
*Russet Norkotah 296*	*0.21*	*COTX04193s-2R/Y*	*1.00*	*NDTX091886-3P/P*	*0.17*	*NDTX4828-2R*	*1.85*
*TXNS106*	*0.21*	*COTX01403-4R/Y*	*0.98*	*TX1617-1W/Y*	*0.17*	*TX12474-1P/R*	*1.83*
*TXNS118*	*0.21*	*NDTX5003-2R*	*0.97*	*AOTX95265-2ARu*	*0.16*	*AOTX95265-2ARu*	*1.82*
*TXNS249*	*0.21*	*NDTX5067-2R*	*0.97*	*NDTX081648CB-4W*	*0.16*	*TX11461-2W*	*1.78*
*ATX9332-8Ru*	*0.20*	*NDTX050169-1R*	*0.82*	*ATX9202-3Ru*	*0.15*	*Rio Rojo*	*1.74*
*AOR07781-2*	*0.16*			*ATX08181-5Y/Y*	*0.13*		

Clones with bold font represent having highest GEBVs; clones underlined are reference varieties; and clones with italic font are the clones with lowest GEBVs.

**Table 7 plants-13-00616-t007:** Genomic-estimated breeding values (GEBV) for potato internal tuber-defect traits (highest—bold font, lowest—italic font and reference varieties—underlined) evaluated in Texas across environments: Dalhart (2019 and 2020) and Springlake (2020 and 2021).

Clone	Percent Hollow Heart	Clone	Percent Vascular Discoloration	Clone	Percent Internal Heat Necrosis	Clone	Percent Internal Defects
**TX1475-3W**	**10.68**	**TX09403-15W**	**5.00**	** Atlantic **	** 1.37 **	**TX1475-3W**	**14.13**
**NDTX8773-4Ru**	**9.12**	**COTX90046-1W**	**4.99**	**COTX10080-2Ru**	**1.18**	**NDTX8773-4Ru**	**12.33**
**ATX84706-2Ru**	**6.80**	**TX09403-21W**	**4.97**	**TX09396-1W**	**1.16**	**ATTX00289-6Y/Y**	**11.37**
**TX13590-9Ru**	**6.46**	**ATTX95490-2W**	**4.75**	**TX05249-11W**	**1.11**	**ATX84706-2Ru**	**11.15**
**TXA549-1Ru**	**6.44**	**COTX90046-5W**	**4.53**	**TX03196-1W**	**0.96**	**TXA549-1Ru**	**10.62**
**AOTX98152-3Ru**	**6.39**	**ATX08181-5Y/Y**	**4.52**	**TX09403-15W**	**0.95**	**TX13590-9Ru**	**10.61**
**ATTX00289-6Y/Y**	**5.01**	**ATTX05186-2R**	**4.31**	**TX09403-21W**	**0.95**	** Atlantic **	** 10.58 **
**COTX09089-1Ru**	**4.44**	**ATX05186-1R**	**4.26**	**TXA549-1Ru**	**0.94**	**AOTX98152-3Ru**	**10.56**
**TX09396-1W**	**4.43**	**White LaSoda**	**4.12**	**AOTX98152-3Ru**	**0.93**	**ATTX05186-2R**	**10.43**
** TXYG79 **	** 4.34 **	**TX05249-10W**	**4.11**	**TX13590-9Ru**	**0.93**	**ATX05186-1R**	**10.38**
Atlantic	2.83	Atlantic	3.92	Russet Burbank	0.58	TXYG79	7.65
Russet Norkotah	1.41	TXYG79	3.01	TXYG79	0.57	Russet Burbank	4.98
Russet Burbank	0.95	Russet Burbank	2.95	Russet Norkotah	0.41	Russet Norkotah	4.53
*NDTX071258B-1R*	*0.24*	Russet Norkotah	2.53	*COTX08365F-3P/P*	*0.31*	*COTX02293-4R*	*1.91*
*ATTX10265-4R/Y*	*0.22*	*AOR07781-2*	*2.43*	*ATTX98453-11Br*	*0.30*	*COTX08365F-3P/P*	*1.90*
*NDTX050184s-1R/Y*	*0.22*	*ATTX98493-2P/P*	*2.43*	*COTX05211-4R*	*0.30*	*TX12474-1P/R*	*1.68*
*NDTX91068-11R*	*0.21*	*TX12474-1P/R*	*2.42*	*COTX05211-5R*	*0.30*	*NDTX071258B-1R*	*1.59*
*NDTX059761-1R/R*	*0.16*	*AOTX03187-1Ru*	*2.41*	*NDTX4271-5R*	*0.30*	*COTX94218-1R*	*1.46*
*COTX00104-7R*	*0.11*	*COTX03187-1W*	*2.38*	*NDTX4784-7R*	*0.30*	*NDTX059775-1W*	*1.44*
*NDTX059828-2W*	*0.11*	*ATX9332-8Ru*	*2.36*	*ATTX06246-1R*	*0.27*	*NDTX059761-1R/R*	*1.33*
*ATTX98493-2P/P*	*0.10*	*COTX02293-4R*	*2.28*	*NDTX5003-2R*	*0.27*	*NDTX050169-1R*	*1.23*
*NDTX059775-1W*	*0.02*	*ATX9312-1Ru*	*2.27*	*NDTX5067-2R*	*0.27*	*NDTX059828-2W*	*0.89*
*AORTX09037-1W/Y*	*−0.06*	*AOTX02136-1Ru*	*2.21*	*NDTX050169-1R*	*0.25*	*ATTX98493-2P/P*	*0.54*
		*COTX08365F-3P/P*	*2.14*				

Clones with bold font represent having highest GEBVs; clones underlined are reference varieties; and clones with italic font are the clones with lowest GEBVs.

## Data Availability

The datasets utilized in this study’s analysis are available upon reasonable request.

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
