# Peer review of "Genetic Basis of Potato Tuber Defects and Identification of Heat-Tolerant Clones"

_plants, 2024, doi:10.3390/plants13050616_

Round 1
Reviewer 1 Report
Comments and Suggestions for Authors
Gautam et al.'s manuscript, entitled "Genetic basis of potato tuber defects and identification of heat-tolerant clones,” discussed how to overcome heat stress, which causes yield losses and quality traits, including external and internal tuber defects, reductions in specific gravity, and increases in reducing sugars that result in suboptimal (darker) processed products (french fries and chips) in potatoes.
The authors studied the panel of 217 potato clones with reference varieties Atlantic Russet, Burbank, Russet Norkotah, and Yukon Gold strains and evaluated them for yield and quality attributes in two seasons at two places: Dalhart (2019 and 2020) and Springlake (2020 and 2021), Texas, and genotyped and phenotyped them. They identified QTLs on chromosomes 1, 3, 4, 6, 8, and 11 for external defects and on chromosomes 1, 2, 3, 10, and 11 for internal defects, and yield-related quantitative trait loci were detected on chromosomes 1, 6, 10.
The authors were well-written and presented the results and discussion nicely. To improve the manuscript, please follow the following suggestions.
Figures 1 and 2 are better shifted to supplementary data.
Line no. 136: Please abbreviate ‘PVY’
Line no. 93-97: better to abbreviate genes
Line 304: Please check the table number.
Line 320: Please check the figure number.
Comments on the Quality of English LanguageEnglish quality good.
Author Response
Thank you for reviewing our paper. We have incorporated your suggestions.
Please see the attachment.

Reviewer 2 Report
Comments and Suggestions for Authors
The paper describes the identification of quantitative trait loci for potato defects. The methods for sample preparation and analyses were described well however I do not see any "highly significant" QTLs regions that provide a strong genetic basis for the various phenotypes described in this study.
Perhaps the author can present the results more concisely -- a figure with co-localised QTLs could be a better way to present multiple low-effect QTLs.
The discussion section is too long simply because there were lengthy discussions of putative genes associated with QTLs when these QTLs are too minor. The authors should consider moving all the minor QTLs to the appendices section and focus on the main large-effect QTLs.
Are the samples too similar hence the low genetic variability?
Comments on the Quality of English LanguageThe quality of English used in this paper is satisfactory.
Author Response
Thank you for reviewing our paper. Your suggestions have been incorporated in the revised version.
Please see the attachment.
